# Circular RNAs Acting as miRNAs’ Sponges and Their Roles in Stem Cells

**DOI:** 10.3390/jcm11102909

**Published:** 2022-05-20

**Authors:** Juan Xiao, Shija Joseph, Mengwei Xia, Feng Teng, Xuejiao Chen, Rufeng Huang, Lihong Zhai, Wenbin Deng

**Affiliations:** 1School of Basic Medicine, Hubei University of Arts and Science, Xiangyang 441053, China; xiaojuan@bjmu.edu.cn (J.X.); 2017168408@hbuas.edu.cn (S.J.); 2020168172@hbuas.edu.cn (M.X.); tengfeng1217@webmail.hzau.edu.cn (F.T.); xuejiao.chen@hbuas.edu.cn (X.C.); 2020168164@hbuas.edu.cn (R.H.); 2School of Pharmaceutical Sciences (Shenzhen), Sun Yat-sen University, Shenzhen 510060, China; 3Jiangxi Deshang Pharmaceutical Co., Ltd., Zhangshu 336000, China

**Keywords:** circular RNAs (circRNAs), gene regulation, stem cells

## Abstract

Circular RNAs (circRNAs), a novel type of endogenous RNAs, have become a subject of intensive research. It has been found that circRNAs are important players in cell differentiation and tissue homeostasis, as well as disease development. Moreover, the expression of circRNAs is usually not correlated with their parental gene expression, indicating that they are not only a steady-state by-product of mRNA splicing but a product of variable splicing under novel regulation. Sequence conservation analysis has also demonstrated that circRNAs have important non-coding functions. CircRNAs exist as a covalently closed loop form in mammalian cells, where they regulate cellular transcription and translation processes. CircRNAs are built from pre-messenger RNAs, and their biogenesis involves back-splicing, which is catalyzed by spliceosomes. The splicing reaction gives rise to three different types of intronic, exotic and exon–intron circular RNAs. Due to higher nuclease stability and longer half lives in cells, circRNAs are more stable than linear RNAs and have enormous clinical advantage for use as diagnostic and therapeutic biomarkers for disease. In recent years, it has been reported that circRNAs in stem cells play a crucial role in stem cell function. In this article, we reviewed the general feature of circRNAs and the distinct roles of circRNAs in stem cell biology, including regulation of stem cell self-renewal and differentiation. CircRNAs have shown unique expression profiles during differentiation of stem cells and could serve as promising biomarkers of these cells. As circRNAs play pivotal roles in stem cell regulation as well as the development and progression of various diseases, we also discuss opportunities and challenges of circRNA-based treatment strategies in future effective therapies for promising clinical applications.

## 1. Introduction

As a dazzling new star of the RNA family, circular RNAs (circRNAs) have attracted much attention in the research because of their newly identified function in cell physiology and disease progression. CircRNAs are widely distributed in cells, and show tissue-specific and developmental-specific expression. In addition, circRNAs are characterized by structural stability, conservation and high intracellular abundance. Important biological functions of circRNAs, such as microRNA (miRNA) sponge phagocytosis, cell cycle regulation, intercellular communication, transcription regulation, translation regulation, disease diagnosis and therapeutic potential, have been revealed.

CircRNAs are a class of special RNAs that covalently form a closed loop in eukaryotes [1]. CircRNAs were first discovered in the 1970s in plant viroids as single-stranded closed RNA [2]. However, they were largely ignored until a few years later, when circRNAs were observed in the cytoplasmic fraction of eukaryotic cell lines using electron microscopy. They were considered as the ‘trash’ RNAs produced by splicing events [3]. Benefiting from the development of bioinformatic tools and high-throughput RNA-sequencing, researchers have found that production of circRNAs is a general feature of the human transcriptome and is ubiquitous in eukaryotes [4,5,6]. Intriguingly, Reut et al. demonstrated that the production rate of circRNAs is mainly determined by intronic sequences, and circularization and splicing compete against each other. For example, the second exon of the splicing factor muscleblind (MBL/MBNL1) is circularized to circMbl, and its flanking introns contain conserved MBL binding sites [7]. As another alternative splicing factor, Quaking (QKI) reguates the production of over one-third of abundant circRNAs and also depends on intronic QKI binding motifs [8]. Therfore, although the human genome has almost 25,000 coding genes, its final products far exceed this number. Seemlingly, circRNAs are “by-products” of pre-mRNA; nevertheless, the mutual restriction and regulation network between pre-mRNA and circRNA have a lot of secrets that need to be continuously explored.

In recent years, intensive scientific investigation has inevitably led to the conclusion that circRNAs have multiple functions such as serving as microRNA sponges, protein scaffolds and being translated into functional polypeptides [4,9]. CircRNAs have a larger half-life than other types of RNA, which makes them potential candidates to serve as diagnostic biomarkers and therapeutic targets. For instance, hsa_circ_002059 and hsa_circ_0001649 were regarded as a potential novel biomarker for the diagnosis of gastric cancer [10] and hepatocellular carcinoma [11] separately. Lei et al. reviewed the expression pattern of circRNAs in different cancers, and these circRNAs might have potential for development as convenient biomarkers for cancer screening [12]. Remarkably, Memczak and colleagues observed that circRNAs were reproducibly and easily detected in clinical standard blood samples, suggesting that circRNAs may represent a new class of biomarkers for human disease [13]. Many studies have discovered that circRNAs play a significant role not only in normal cell, tissue and organ development, but also in disease pathogenesis [13]. For example, CDR1as, also called CiRS-7, is a superstar molecule. The overexpression of CDR1as significantly promotes insulin biosynthesis and secretion through the CDR1as/miR-7 axis, which provides a potential target for improving b cell function in diabetic patients [14]. Further studies discovered that CDR1as also functions in prion diseases, Alzheimer disease and Parkinson’s disease [15,16,17]. Moreover, CDR1as/miR-7 axis may also perform a crucial role in cancer-associated pathways [3], and many circRNA/miRNA axis functions in cancer, as shown in Table 1.

Recently, the crucial roles of circRNAs in various types of stem cells were discovered [18,19,20]. This review provides an overview of the biogenesis and functions of circRNAs, highlights the mechanisms of action of circRNAs in stem cells, and discusses the potential clinical applications of circRNAs.

## 2. Biogenesis and Characteristics of circRNAs

### 2.1. Biogenesis and Types of circRNAs

CircRNAs are the products of back-splicing. Through the back-splicing mechanism, circRNAs are generated through pre-mRNA splicing with spliceosomal splicing machinery acting as the catalyst. The spliceosomal machinery forms the canonical splice signs assisting in the removal of intronic sequence to form CircRNA [7,56,57]. The role of spliceosomal machinery in catalysation of back-splicing was proved by using the isoginkgetin treatment, which inhibits canonical spliceosome, causing a reduction of both circRNAs and linear transcripts [57]. Additionally, some enzymes such as ribozymes I and II may be involved in the catalysation of back-splicing, though the mechanism of how it catalyzes the reaction is not completely clear [58]. There is the involvement of the complementary flanking element in the formation of circular RNA; the complementary material is likely to be found in the structure of intron, where they convey the splice site together to form the circle [59,60,61]. The ALU element is the common complementary element involved in the process [62]. Additionally, there is the involvement of proteins in the regulation of circular RNA biogenesis; proteins do so by creating an environment for exon circularization. For example, the muscle blind protein binds to the intron, forming a gap in between to initiate exon circularization to form circular RNA [7]. Also, the quaking protein (QKI) is another one involved in the regulation of circular RNA biogenesis. The QKI binds to flanking intron to form a loop, which enhances circularization [8]. In some studies, it was shown that apoptotic endonuclease G (EndoG) can produce splice-switching oligonucleotides that act very similarly to circRNA. However, such nucleotides were disrobed for human Telomerase Reverse Transcriptase (hTERT) and Deoxiribonuclease I (DNase I) [63,64,65,66]. Whether EndoG plays a role in the production of circRNAs or not needs to be further studied.

Based on how they are formed, circular RNAs are divided into three types (Figure 1): the intronic circular RNA (ciRNA), the exonic circular RNA (EcircRNA), and the exon–intron circular RNA (EIcircRNA). The exonic circular RNA is formed by an exon skipping event that forms the lariat, which contains an exon assisting the removal of intron sequence by internal splicing to form the known exon circRNA [67]. Intronic circular RNA (ciRNA) is formed from intron lariats that escape degradation by de-branching enzymes [58,68]. The mechanism of EIcircRNA formation is the same as that of EcircRNA formation, but there is intron retention allowing the circle to contain both exon and intron.

### 2.2. Cellular Localization of circRNAs

CircRNA is mostly cytoplasmic and can be transported to the cytosol [5,62]. According to the length of the circRNAs, they are sorted by the nuclear export machinery and their localization is controlled by UAP56 and URH49, which are responsible for the export of long (>1200 nt) and short circRNAs (<400 nt), separately [69]. However, ciRNAs or EIciRNAs are restricted in the nucleus in human cells [70,71]. Nuclear-retained circRNAs functioned in transcription regulation. It was found that EIciRNAs could interact with U1snRNPs and the complexes associate with PolII at the promoters of their host genes to enhance gene expression [71].

### 2.3. Abundance, Stability, Multiplicity and Specificity of circRNAs

Based on recent studies, circRNAs have some distinct features from other RNAs: (i) Abundance: In fibroblasts, 14% of the transcribed genes are composed of circRNAs, and the expression level is much greater than that of their host RNAs [62]; (ii) Stability: Compared to linear mRNAs, the lack of 5′ caps and 3′ poly (A) tails make circRNAs avoid RNase R digestion and longer half-lives make circRNAs naturally more stable [5]; (iii) Multiplicity: Different genes can produce circRNAs with different combinations of exons and introns due to alternative splicing; (iv) Specificity and Conservation: The expression pattern of circRNAs has tissue-specificity, and the data showed that circRNAs expressed most highly and specifically in the human brain, and secondly in the liver and heart as compared to other tissues [72]. On the other hand, circRNAs may have higher sequence conservation than other types of RNA in mammals [72]. These features of circRNAs imply their diverse potential biological functions and clinical applications.

## 3. Function of circRNAs

### 3.1. Transcriptional Regulation

#### 3.1.1. miRNA Sponges

miRNAs have been shown to function as a gene expression regulator in some diseases [73]. They have a post-transcriptional regulate gene expression that inhibits translation or facilitates degradation through direct base pairing to the 3′-untranslated region (UTR), the coding region or the 5′-UTR of target mRNA [74]. CircRNAs can function as miRNA sponges and act as competitive endogenous RNA to deregulate mRNA by miRNA [75]. ciRS-7 or CDR1as (sponge for miR-7) is the best characterized circRNA produced from vertebrate cerebellar degeneration-related 1 (CDR1) antisense transcript preferentially expressed in the human and mouse brain. It binds to miRNA and regulates its function [3]. There are more than 70 miR-7 binding sites on ciRS-7, limiting the influence of miR-7 on target mRNAs [2]. Another is circSry, which is encoded by the sex-determining region Y (SRY) gene, and encompasses 16 miR-138 binding sites [21]. Different circRNAs generated from the cattle casein (CSN) gene, which are highly expressed in the bovine mammary gland, can sponge the miR-2284 family, which targets CSNIS1 and CSN2 mRNAs [76]. Table 1 summarize some of different kinds of circRNA that can form miRNA sponge. As our understanding of circRNA grows, other types of circRNA that can form miRNA sponge are likely to arise.

#### 3.1.2. Regulation of Alternative Splicing

CircRNA biogenesis appeared to compete with splicing. The MBL protein regulates alternative splicing, and in the process, they bind with circMBL generated from pre-mRNA circularization to form the MBL–MBL interaction which compete with canonical splicing machinery [7]. QKI protein appeared to regulate pre-mRNA splicing as the MBL [77]; however, its mechanism is not clearly known.

### 3.2. Translating to Peptide

CircRNAs are capable of undergoing translation though they lack the necessary essential element for cap-independent translation such as the 5′cap and poly (A) tail [78]. However, following the incorporation of m6A RNA modification in the 5′untranslated region (UTR) of circRNAs can induce translation [79]. Circ-ZNF609 encodes a protein through IRES (internal ribosome entry sites) active in the UTR Circ-ZNF609 [80]. CircMBL can be translated in a cap-independent way, and circMbl1-encoded protein is enriched in synaptosomes and modulated by starvation and FOXO [81]. These two studies have opened a new research direction, and more and more researchers have begun to pay attention to the translational function of circRNAs. Recently, Zhang et al. found that circPPP1R12A-73aa encoded by circPPP1R12A promoted tumor pathogenesis and metastasis of colon cancer by activating the Hippo–YAP signaling pathway [82]. Li et al., discovered that circ-HER2 encoded a novel protein, HER2-103; however, Circ-HER2/HER2-103 positive TNBC (Triple negative breast cancer) patients harbored a worse overall prognosis than circ-HER2/HER2-103 negative patients [83]. Additionally, a novel protein AXIN1-295aa encoded by circAXIN1 to promote gastric cancer progression activates the Wnt/β-catenin signaling pathway [84]. Currently, the function of the truncated peptides encoded by circRNAs are mostly similar to the full-length protein counterparts (circFBXW7-185 aa [85,86]). However, some peptides originating from circRNAs exert functions independent of or even opposed to those of their host gene products (circFNDC3B-218 aa [87]). All these studies provide strong support for a clearer understanding of the human proteome. More studies should be carried out to discover the regulatory mechanisms and processes of circRNAs translation.

### 3.3. Function as Protein Scaffolds, Decoys and Recruiters through Interaction with Proteins

Emerging studies have shown that a batch of circRNAs can serve as protein decoys, scaffolds and recruiters in diverse physiological and pathological contexts. Zhou et al. reviewed 80 circRNA–protein interaction relationships classified by manners of action [88], such as altering interactions between proteins, blocking proteins from DNA or RNA or proteins, recruiting transcription factors to chromatin, recruiting modifying enzymes to chromatin, recruiting chromatin remodelers, ternary complexes regulating RNA stability, ternary complexes regulating translation, translocating proteins to the nucleus, and translocating proteins to the cytoplasm. CircFoxo3 plays important roles both in altering interactions between proteins (P53 and MDM2 [89]; p21 and CDK2 [90]); and translocating proteins to the cytoplasm (ID-1, E2F1, FAK, HIL1α [91]) manners. Furthermore, a recent study demonstrated that circRNAs can interact with proteins and then promote their nuclear translocation. Circ-DNMT1 can enhance the nuclear translocation of P53 and AUF1, and then induce cellular autophagy and enhance breast cancer cell proliferation separately [92].

Therefore, the same circRNA can interact with different proteins in diverse manners, and different circRNAs can also interact with the same proteins in the same or distinct manners, and then play crucial roles in various biological processes. In the future, more studies will reveal the interaction relationship between circRNAs and proteins.

## 4. Roles of circRNAs in Stem Cells

There is a day-to-day in-depth understanding of circRNAs among researchers associated with the improvement of biological technologies. CircRNAs are a stable, diverse and conserved class of RNA molecules, representing a novel type of regulatory ‘non-coding’ RNA that can act in gene regulation by acting as a miRNA sponge, regulating transcription and translation, and pre-mRNAs splicing and translational function. The study of this RNA breaks new ground in medical science; it expands our understanding with different new diagnosis measures of disease and treatments. According to Tissue-Specific CircRNA Database (TSCD) (http://gb.whu.edu.cn/TSCD, accessed on 19 May 2022), circRNAs have the highest and most specific expressions in the human brain, followed by liver and heart compared to other tissues [88]. Therefore, the more complex the organ or biological process is, more are the circRNAs needed for regulation; that is to say, circRNAs may be one of the most delicate gene-regulated molecules. The differentiation and proliferation of stem cells is an extremely complex biological process, and circRNAs may play a pivotal role in this process.

Stem cells are cells found in the body. They are undifferentiated cells commonly found in the skin, brain, marrow, skeletal muscles and liver cells, and their differentiation gives rise to several types of different cells in the body of the organism. The study of stem cells brings great hope in medicine as it opens a way to treat even grave diseases that were once incurable. To date, the existing research on circRNA function on stem cells is quite limited; however, circRNAs in stem cells have attracted considerable attention for their abundance in expression specificity and roles in promising clinical applications. CircRNAs and their host genes generally have cells and tissues specificity, but they may function in the same and different tissues [93]. On the other hand, the number of circRNAs detected is significantly more than that of genes [94]. Therefore, there is reason to believe that circRNAs and their host genes play crucial roles in different stem cells in similar or different manners, and furthermore, the regulatory relationship between circRNAs and their host genes is also extremely complex. Revealing these will bring revolutionary value to stem cell research, thereby identifying new targets of circRNA-related therapeutics and creating new strategies for improving stem cell therapy [95].

Originally, circRNAs were found to be involved in regulation of stem cells mostly by high-throughput sequencing. For instance, Zheng et al. discovered the distinct expression pattern of circRNAs in periodontal ligament stem cells (PDLSCs) during osteogenesis, which suggests the relationship and high involvement of circRNA in bone disorder [96]. Sun et al. suggested that upregulation of circRNA may repair the damaged endometrium by Wharton’s jelly-derived mesenchymal stem cells (WJ-MSCs) [97]. In the study, it was seen that 7757 circRNAs were differentially expressed in ESCs from the cocultured group; they also found that hsa_circRNA_0111659 may bind with miR-17-5p, miR-20b-5p and miR-93-5p, and function in the process of repair by WJ-MSCs. Furthermore, it was discovered that the expression of circRNAs in human placental chorionic plate-derived mesenchymal stem cells was pretreated with hypoxia. The study identifies 85 circRNAs as upregulated, 17 as downregulated, and 27 as upregulated by more than two-fold, suggesting that the expressed circRNAs may be involved in different biological processes and molecular regulatory mechanisms of hpcpMSC [98]. Lei et al. identified 5602 circRNAs in human-induced pluripotent stem cells (hiPSCs) and hiPSC-derived cardiomyocytes (hiPSC-CMs), and then suggested that circSLC8A1, circCACNA1D and circSPHKAP may serve as novel biomarkers of the cardiogenesis of hiPSCs [99]. Other profiling studies for human circRNAs expression in various stem cells have emerged, such as human dental pulp stem cells (hDPSCs) [100] and hematopoietic stem cells (HSCs) [101].

Subsequently, numerous studies about specific circRNAs and their concrete regulatory signaling pathways by sponging miRNAs have been carried out. We summarized the studies about different circRNAs’ functions in different types of stem cells as follows. Table 2 and Figure 2 show some specific functions of different circRNAs in multiple stem cells.

Pluripotent stem cells. This type of stem cells includes embryonic stem cells (ESCs) and induced pluripotent stem cells (iPSCs). Some studies have revealed that circRNAs exert significant and comprehensive effects on ESCs, including the modulation of human pluripotency and differentiation. For example, circBIRC6 was upregulated in undifferentiated hESCs by binding with miR-145 and miR-34a [102]. Some studies showed circSLC8A1 function in hiPS cell-derived cardiomyocytes through binding with miR-133a. Furthermore, the over-expression of circSLC8A1 could indicate pathological status in heart disease [126]. Downregulated circITCH in doxorubicin-treated human iPSC-derived cardiomyocytes can upregulate SIRT6, Survivin and SERCA2a by sponging miR-330-5p and then alleviate DOX-induced cardiomyocyte injury [103]. A better understanding of circRNAs in the context of iPSCs would definitely benefit future biomedicines aiming to utilize iPSCs in the clinic.

Somatic stem cells. Somatic stem cells (SSCs) are found in unique stem cell niches located in different adult tissues. Li et al. suggested the regulatory role of circRNA in the osteoblastic differentiation of periodontal ligament stem cells via the miR-7/GDF5/SMAD and p38 MAPK signaling pathway, which found significant upregulation of circRNA (CDRI) during the process of osteogenic differentiation [127]. Table 2 has listed majority research about circRNA function in different types of SSCs.

Otherwise, a number of circRNAs were considered as novel and outstanding biomarkers of SSC differentiation. CircFOXP1 is a marker of undifferentiated MSCs and was able to preserve the MSC multipotent state by sponging multiple miRNAs [20]. Further studies indicated that circFOXP1 sponges miR-33a-5p and miR-30a-3p to upregulate *FOXP1* and *RUNX2*, *ALP* and *OCN*, thus facilitating osteogenesis in hADSCs [109,128]. Notably, numerous studies on reducing the injury caused by disease by regulating the circRNA–miRNA–gene pathway derived from stem cells exosomes were carried out. For example, based on the circRNA_0031672/miR-21-5p/PDCD4 pathway, miR-21-5p-expressing BMSCs could alleviate myocardial ischemia/reperfusion injury [129]. circDIDO1 derived from MSCs could suppress hepatic stellate cell activation through miR-141-3p/PTEN/AKT pathway in human liver fibrosis [130]. Circ-Fryl derived from exosomes of ADSCs could attenuate sepsis-induced lung injury through the miR-490-3p/SIRT3 pathway [131]. These will provide new ideas and targets for stem cell therapy.

Cancer stem cells. Cancer stem cells (CSCs) are regarded as a sustainable source of malignant cells. These cells may serve as the culprit contributing to drug resistance, tumor recurrence, metastasis and progression. In addition to functioning as miRNA sponges listed in Table 2, circRNAs also function as other forms. For instance, circGprc5a can be translated into a new short peptide which binds with Gprc5a and increases its function to drive the self-renewal of bladder CSCs [132]. Cirs-7 can facilitate the cytoplasmic localization of NF-Κb and then regulate downstream genes to influence the apoptosis of ovarian cancer stem cells (OCSCs) [133,134]. In addition, some circRNAs play important roles in CSCs by interacting with other proteins. For example, circCTIC1 interacts with NURF complex and drives the self-renewal of colon TICs [135]. CircZKSCAN1 blocks FMRP binding to CCAR1 mRNA and negatively modulates CSCs [136]. Certainly, research on circRNAs in CSCs is still in the initial stage [137]. However, it is believed that circRNA is a dazzling star in the diagnosis and treatment of cancer.

It has been reported that the dynamics of circRNA production remained largely unchanged in both human ES cells and human ES-cell-derived forebrain neurons, but the abundance and diversity of the circRNAs in forebrains increased substantially [93]. The high expression of circRNAs in neurons may be attributed to their stability and, therefore, accumulation in these slow-dividing cells. Many circRNAs become more abundant during synaptic formation, and these observations suggest that circRNA may regulate synaptic function. Among humans, mice and fruit flies, some brain-specific circRNAs are conserved, suggesting that they have a role in neuronal functions, possibly associated with memory due to their high stability. Excitatory synaptic transmission dysfunction associated with neuropsychiatric disorders also supports the importance of circRNAs in neurons in mouse models of psychiatric disorders.

## 5. Opportunities and Challenges of circRNA-Based Therapeutic Technology

CircRNAs have shown unique expression profiles during differentiation of stem cells and could serve as promising biomarkers of these cells. Therefore, circRNAs play pivotal roles in stem cell regulation as well as in the development and progression of various dis-eases, and may have promising therapeutic potential. The circRNA technology may overcome the limitations of the mRNA technology on multiple fronts. Once the circRNA technology is more developed and mature, it will be able to translate into various types of proteins and perform the same function as mRNAs. Therefore, it has great potential to become the next generation of RNA-based therapy. Although the mRNA technology is currently sought after, potential problems include limited stability, expression without tissue specificity and existence of immunogenicity. For all the limitations of mRNA technology, the “dark horse” circRNA technology is expected to crack one by one.

### 5.1. CircRNAs Are More Stable Than mRNAs Because of Their Circle Morphology

Compared to linear mRNAs, the lack of 5′ caps and 3′ poly(A) tails make circRNAs resistant to RNase R and naturally more stable [5]. The instability of mRNAs originates from the existence of its terminal. The ubiquitous RNase in the environment brings great challenges to the research, production, preparation and storage of mRNAs. The messenger role of the Central Dogma also determines that the half-life of mRNAs in the body is often very short. CircRNAs have a natural advantage in morphology. Because the structure is a circle without an end, it is not easily recognized by the body’s RNA degradation system, which makes circRNAs more stable in the body than mRNAs, and circRNAs have a longer window of time to exert their effects. Therefore, circRNAs can extend the frequency of administration by taking advantage of the ability to continuously express the proteins. If mRNAs need to be injected every two or three days, it is possible that circRNAs can be stable for several months after injection.

### 5.2. CircRNAs Are Tissue Specific and Have Less Side Effects Than mRNAs

In terms of tissue specificity, since mRNAs are a core molecule in the middle of the Central Dogma, all cells express mRNAs, making mRNAs less tissue specific. CircRNAs can be designed and optimized to achieve specific tissue expression. The tissue specificity of circRNAs will be a huge advantage in controlling potential side effects. Once we achieve high expression only in specific target cells, even if potential delivery problems lead to entry into non-targeted cells, specific expression can reduce the side effects of circRNAs in those other cells.

In addition, in terms of immunogenicity, mRNAs have a strong immune response, which is a double-edged sword. On the one hand, the innate immune response activated by mRNAs promotes subsequent antigen presentation; on the other hand, overactivation of innate immunity can produce a severe immune response, leading to symptoms similar to viral infection, triggering side effects such as inflammation and even autoimmune diseases. Therefore, base modification is often needed to reduce the immunogenicity of mRNAs. In contrast, circRNAs will have low immunogenicity even without base modification.

The mRNA-based nucleic acid vaccine has made great progress in the global fight against the pandemic, but there is still a lot of room for improvement in effective duration and safety. Recently, a Chinese team successfully developed a platform for the production of “circular RNA vaccine”. The researchers hypothesized that unlike current DNA vaccines, which use linear mRNA molecules to produce antigens, the circular structure of circRNAs would be more stable in the body. Experimental data from mouse and rhesus monkey models strongly support the idea that the coronavirus S protein antigen produced by the circular RNA vaccine has a longer half-life in laboratory animals than the linear mRNA vaccine and is effective in triggering an immune response [138]. At the same time, the circular RNA vaccine made with the template of Delta strain showed good protection against Alpha, Beta, Delta and Omicron strains. The future performance of the circRNA vaccine in clinical trials is worth looking forward to.

### 5.3. Challenges of circRNA Technology

Nonetheless, circRNA technology still has a long way to go. At present, the research on circRNAs is at an early stage, and there are still technical difficulties to explore in this field. First of all, because it is a young technology, there are a lot of uncertainties about the practical application of circRNA therapy. The successful launch of a circRNA drug involves early development, clinical trials, production, transportation, etc. So far, many of these areas in the circRNA therapeutics field are still unknown, and need further verification. Another difficulty is the nucleic acid drug delivery problem. Exactly what kind of delivery system is the best solution for circRNAs? All of these problems need to be continually explored and solved.

## 6. Conclusions and Perspectives

Mounting evidence has shown that the upregulation of circRNAs and downregulation of circRNAs in stem cells can offer a significant advantage in stem cell therapy, which is very important therapy in current days. However, the emerging roles of circRNAs in diverse stem cells remain largely unknown. Further research should be carried out on circRNAs and their mechanism in diverse stem cells. Meanwhile, the role of circR-NAs as a novel biomarkers of stem cell differentiation also needs more attention. There-fore, continuous and in-depth study of circRNAs is needed to provide important clues which will be helpful in medical science and human life in general. For example, the mechanisms underlying the biogenesis and biological function of circRNAs, the regulatory relationship of circRNAs and the host genes, the research methodologies of circRNAs, degradation mechanism of circRNAs, the clinical application of circRNAs, etc.

According to the features of circRNAs, circRNA technology may have more advantages than mRNA technology. For the future development of circRNA technology, we need to build a systematic R & D platform based on the advantages of circRNAs, explore more effective therapeutic models, and further build the next generation RNA-based therapeutic platform. It is known that the cyclization of mRNAs presents an obvious problem. There is a “small tail” left after cyclization, called a “scar sequence”, which is added to the mRNAs to cyclize, but cannot be removed. However, this extra sequence can cause some side effects and make it difficult to avoid scar sequence in design. For the synthesis or preparation of circRNAs, there is a need to develop a sequence-free, efficient, easy-to-scale-up optimal cyclization technology. In terms of efficient translation, circular RNA achieves far more than the cumulative translation of linear RNA, and a certain degree of tissue-specific expression. Previous studies have demonstrated the low immunogenicity of circRNAs in mice and the high expression of the target gene.

Looking ahead, the circRNA technology could really do a lot of good things once it matures, and could even become an alternative to many existing therapies. In addition, many rare diseases are actually caused by the loss of certain proteins, particularly if the protein is very large. While expressing large proteins is difficult for existing therapies, which may be difficult for gene therapy, circRNAs could overcome the barrier and provide viable treatments for diseases that have no effective therapies.

## Figures and Tables

**Figure 1 jcm-11-02909-f001:**
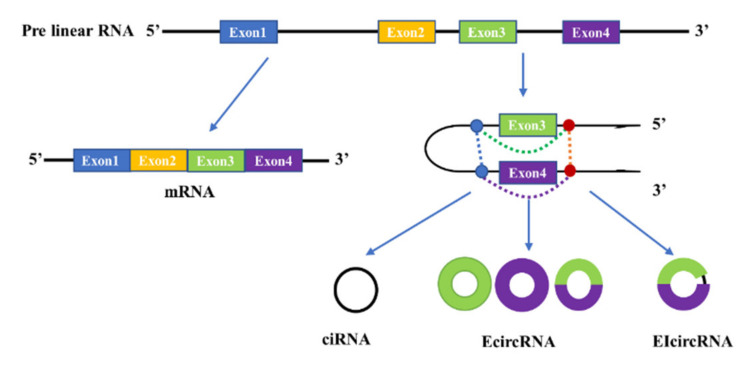
The proposed models of circRNAs biosynthesis. The mature mRNA containing all exons and the three types of circRNAs produced by back-splicing possibly occurred in Exon3 and Exon4. Intronic circular RNA (ciRNA) only includes the sequence of Intron3, and the exonic circular RNA (EcircRNA) may include the sequence of Exon3, Exon4 or mixed sequnece of Exon3 and Exon4. The exon–intron circular RNA (EIcircRNA) includes the sequence of Exon3, Exon4 and Intron3.

**Figure 2 jcm-11-02909-f002:**
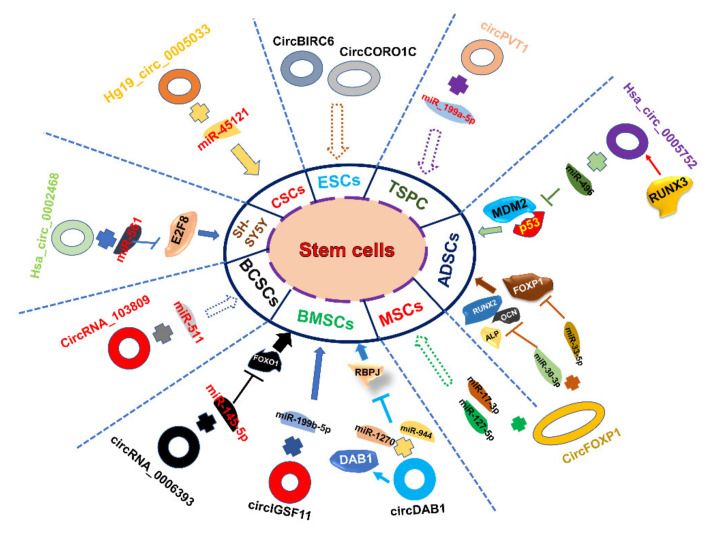
Schematic diagram of the roles and regulatory pathways of partial stem cell-associated circular RNAs. In ESCs, circRNAs (e.g., CircCORO1C, CircBIRC6) regulate the differentiation and pluripotency of hESCs. In TSPC, circPVT1 sponges miR-199a-5p to inhibit TSPC senescence. In ADSCs, hsa_circ_0005752 derived from the RUNX3 gene promotes the osteogenic differentiation of adipose-derived stem cells through binding with miR-496 and then releases the limit of MDM2-p53. CicrFOXP1 can bind with different miRNAs and then regulate the downstream target genes of miRNAs both in ADSCs and MSCs. In BMSCs, circDAB1 (derived from DAB1 gene)/miR-944 and miR-1270/RBPJ axis, circIGSF11/miR-199b-5p and circRNA_0006393/miR-145-5p/FOXO1 axis play essential roles in BMSCs. hsa_circ_0002468/miR-561/E2F8 axis regulates SH-SY5Y differentiation and proliferation. circRNA_103809/miR-511 and hg_circ_0005033/miR-45121 facilitates the migration and invasion of BCSCs and CSCs.

**Table 1 jcm-11-02909-t001:** Some circRNAs act as miRNA sponge and their function.

Name	Organism	miRNAs It Binds	Function	References
ciRs-7/CDR1as	Homo sapien; Mus musculus	miR-7	Prevent downregulation of target gene	[2,3]
circSry	Homo sapien; Mus musculus	miR-138	Involve in cancer pathogenesis; Regulate hypoxia-induced apoptosis in cardiac myocytes	[21,22]
Hsa-circ-000595	Homo sapiens	miR-19a	Decreased apoptosis in human aortic smooth muscle cell	[23]
circTCF25	Homo sapiens	miR-107; miR103a-3p; miR-206	Upregulation of CDK6 in urinary bladder carcinoma tissue Osteosarcoma cell proliferation and migration	[24,25]
circ-ITCH	Homo sapiens	miR-7; miR-214	Osteosarcoma Migration and Invasion promoting linear ITCH expression	[26,27]
circRNA_001569	Homo sapiens	miR-145	Cell proliferation regulation	[28]
circZNF609	Homo sapiens	miR-150; miR-145-5p; miR-150-5p; miR-138-5p; miR-483-3p; miR-186-5p; miR-134-5p; miR-188; miR-1224-3p; miR-342-3p; miR-501-3p; miR-615-5p; miR-22-3p; miR-338-3p; miR-142-3p; miR-623; miR-1224-3p; miR-197-3p	Cancer cell growth, migration and invasion	[29,30,31,32,33,34,35,36,37,38,39,40,41,42,43,44,45,46,47,48,49,50,51]
circZNF91	Homo sapiens	miR-23b-3p;	Chemoresistance of normoxic pancreatic cancer cells by enhancing glycolysis; Epidermal stem cell differentiation	[52,53]
circBFAR	Homo sapiens	miR-34b-5p; miR-513a-3p	Progression of pancreatic ductal adenocarcinoma Proliferation and glycolysis in gastric cancer	[54,55]

**Table 2 jcm-11-02909-t002:** Different circRNAs and their roles in multiple stem cells.

Stem Cell Types	CircRNA	Role in Stem Cells	miRNAs It Binds	References
Pluripotent stem cells	circBIRC6; circCORO1C	Regulates the differentiation and pluripotency of hESCs	miR-34a, miR-145	[102]
	circSLC8A1	Overexpression of circSLC8A1 is related to heart disease from the study of hiPS cell-derived cardiomyocytes	miR-133a	[99]
	CircITCH	Downregulated CircITCH in doxorubicin-treated hiPS cell-derived cardiomyocytes can alleviate DOX-induced cardiomyocyte injury	miR-330-5p	[103]
Somatic stem cells	circRNA-33287	Promotes osteogenic differentiation of maxillary sinus membrane stem cells	miR-214-3p	[104]
	hsa_circ_0002468	Regulates SH-SY5Y differentiation and proliferation	miR-561	[105]
	hsa_circ_0005752	Derived from RUNX3, promotes the osteogenic differentiation of adipose-derived stem cells through release the limit of MDM2-p53	miR-496	[106]
	circFOXP1	Accelerates differentiation and proliferation of MSCs	miR-17–3p and miR-127–5p	[20]
	circRNA_103809	The highest expressed circRNAs identified in BCSCs, promotes the self-renewal, migration and invasion capabilities of bladder cancer	miR-511	[107]
	hsa_circ_0074834	As a ceRNA in bone mesenchymal stem cells (BMSCs)	miR-942-5p	[108]
	circRNA-23525	Regulates osteogenic differentiation of adipose-derived mesenchymal stem cells	miR-30a-3p	[109]
	circPVT1	Inhibits tendon stem/progenitor cell (TSPC) senescence	miR-199a-5p	[110]
	circDAB1	Derived from DAB1 gene to upregulate RBPJ through sponging miRNAs, and upregulates the host gene DAB1 in BMSCs	miR-1270 and miR-944	[111]
	circ—0006393	Promotes osteogenesis in human BMSCs by inducing the FOXO1 gene	miR-145-5p	[112]
	circIGSF11	Inhibits the osteogenic differentiation of hBMSCs	miR-199b-5p	[113]
	circ-0019693	Promotes osteogenic differentiation of BMSCs by inducing PCP4 gene	miR-942-5p	[114]
	circ-011235	Counteracts the harm of irradiation treatment on BMSCs through the miR-741-3p/CDK6 pathway	miR-741-3p	[115]
	circSmg5	Accelerates the osteogenic differentiation of BMSCs through miR-194-5p/Fzd6 and beta-catenin signaling	miR-194-5p	[116]
	circ-0005835	Inhibits NSC proliferation and differentiate to neuron	miR-576-3p	[117]
	circ-0002113	Lacking MSCs suppress myocardial infarction through regulate RUNX1	miR-188-3p	[118]
	CircFAT1	Regulates osteogenic differentiation of periodontal ligament stem cells (PDLSCs) by regulating SMAD5	miR-4781-3p	[119]
Cancer stem cells	circ-ITCH	Promotes the self-renewal and stemness of CSCs by repressing the expression of CTNNBIP1	miR-214	[120]
	hsa_circ_0020397	Promotes malignant proliferation of liver CSCs	miR-138	[121]
	hsa_circ_0005075	Promotes proliferation and differentiation of breast CSCs and a new biomarker of BCSCs	miR-93	[122]
	Circ-008913	Regulates acquisition of CSC-like properties and neoplastic capacity of arsenite-transformed HaCaT cells	miR-889	[123]
	Hg19_circ_0005033	Promotes proliferation, migration, invasion and chemotherapy resistance of CD133+ CD44+ laryngeal CSCs	miR-45121	[124]
	CircPRMT5	Facilitates UCB cell’s EMT and/or aggressiveness	miR30-c	[125]

## Data Availability

Not applicable.

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
