# Peer review of "Circular RNAs Acting as miRNAs’ Sponges and Their Roles in Stem Cells"

_jcm, 2022, doi:10.3390/jcm11102909_

Round 1

Reviewer 1 Report

In this review article by Xiao et al have summarized circular RNAs in stem cells and its therapeutic potential. The authors have tried to summarize the current literature, but it is inadequate to be considered for publication in its present form. Below are the comments regarding this review article:

  • Based on its present form it is not so evident that this review will add valuable contribution to this research field as compared to previous ones (Lu et al. 2022; Wang et al 2020 etc.), in which circRNAs in stem cells were also presented. The authors need to provide a more personal insight to the background of the topic and need to perform thorough literature review which is not adequate in its present format.
  • The major focus of this review article is circRNAs in stem cells but the review article in its current format is not comprehensive enough. Although authors have tried to summarize the existing literature but I would highly recommend to include separate sub sections of circRNAs in stem cell renewal, differentiation of different cell types.
  • The role of circular RNAs in cancer stem cells need to be discussed in this review article.
  • The biological function of circRNAs is inadequately described. There was no information literature review regarding the role of circRNAs in protein scaffolding, protein decoy and protein recruitment.
  • Although the authors have included a subheading of characteristics of circRNAs but there was lack of description about this. The authors should explain the abundance, stability and specificity of circRNA as characteristics. Although they have mentioned in parts about this in the section 'Opportunities and challenges of circRNA-based therapeutic technology' but they should emphasize this into characteristics as well.
  • Figure 2 should include the role of circRNAs in regulating cancer stem cells.

Author Response

In this review article by Xiao et al have summarized circular RNAs in stem cells and its therapeutic potential. The authors have tried to summarize the current literature, but it is inadequate to be considered for publication in its present form. Below are the comments regarding this review article:

Thanks for your suggestions. We have revised our manuscript incorporating all the suggestions of the reviewers.

Q1: Based on its present form it is not so evident that this review will add valuable contribution to this research field as compared to previous ones (Lu et al. 2022; Wang et al 2020 etc.), in which circRNAs in stem cells were also presented. The authors need to provide a more personal insight to the background of the topic and need to perform thorough literature review which is not adequate in its present format.

Response: Thanks for your suggestions. We have cited these relevant reviews about circRNAs in stem cells, and we believe the authors have done a great job summarizing the relevant information on circRNAs in stem cells. Therefore, we have tried to add our own views and the latest research progress in our article.  

Q2: The major focus of this review article is circRNAs in stem cells but the review article in its current format is not comprehensive enough. Although authors have tried to summarize the existing literature but I would highly recommend to include separate sub sections of circRNAs in stem cell renewal, differentiation of different cell types.

Response: Thank you for your advice. We have separated Section 4 to sub-sections on the roles of circRNAs in different stem cell types, and we also rearranged Table 2.

Q3: The role of circular RNAs in cancer stem cells need to be discussed in this review article.

Thanks for this suggestion. We have added the role of circular RNAs in cancer stem cells as a sub-section in Section 4.

Q4: The biological function of circRNAs is inadequately described. There was no information literature review regarding the role of circRNAs in protein scaffolding, protein decoy and protein recruitment.

Response: Thanks for this reminder. We have added more contents of the role of circRNAs in protein scaffolding, protein decoy and protein recruitment in Section 3.3. Furthermore, we changed the subheading of Section 3.2(Translate to peptide) and Section 3.3(Function as protein scaffolds, decoys and recruiters through interaction with proteins).

Q5: Although the authors have included a subheading of characteristics of circRNAs but there was lack of description about this. The authors should explain the abundance, stability and specificity of circRNA as characteristics. Although they have mentioned in parts about this in the section 'Opportunities and challenges of circRNA-based therapeutic technology' but they should emphasize this into characteristics as well.

Response: Thank you for your suggestion. It is necessary to add the content about the abundance, stability and specificity of circRNAs as characteristics in Section 2. Therefore, we added a Section 2.3 (subheading: Abundance, stability, multiplicity and specificity of circRNAs) to explain these features.

Q6: Figure 2 should include the role of circRNAs in regulating cancer stem cells.

Response: Thanks for your advice. We have added the role of circRNAs in regulating cancer stem cells in Figure 2.

Reviewer 2 Report

The manuscript entitled "Circular RNAs in stem cells and therapeutic potential" presents a review of circRNA biogenesis and their roles in stem cells. The manuscript is well constructed, but I have some doubts/suggestions I would like to address:

  1. I believe the therapeutic potential of circRNA is a perspective, more than the focus of the review. The main point of the review is to present the circRNA acting as miRNAs' sponges and their role in stem cell differentiation. Hence, I believe the title of the manuscript should be altered.
  2. In the same logic, the abstract of the manuscript is not presenting the full content of the research. Biomarkers are very interesting in the clinical scenario, but from the data presented, I believe there is not enough data to provide the concept of a circRNA biomarker yet.
  3. Section 5 has only two references cited.
  4. Figure 1 should have a legend fully explaining its content. Figure 2 (which is excellent) should also be more explained.
  5. The sentences in lines 101-105 are confusing. Please rephrase it.

Minor points:

  1. The link for the TSCD database is not working.
  2. MBL protein should be first mentioned in the Introduction section, as QKI.
  3. I suggest the first paragraph of the Conclusions and Perspectives be moved to Section 4.

Author Response

The manuscript entitled "Circular RNAs in stem cells and therapeutic potential" presents a review of circRNA biogenesis and their roles in stem cells. The manuscript is well constructed, but I have some doubts/suggestions I would like to address:

Q1: I believe the therapeutic potential of circRNA is a perspective, more than the focus of the review. The main point of the review is to present the circRNA acting as miRNAs' sponges and their role in stem cell differentiation. Hence, I believe the title of the manuscript should be altered.

Response: Thanks for your specific suggestion. In the study of circRNAs, the biological function of miRNAs 'sponges is well understood, especially in the research of stem cells. However, original research of circRNAs in stem cells focused on transcriptome. Subsequently, numerous studies about specific circRNAs and their concrete regulatory signaling pathways by sponging miRNAs have been carried out. We believe there will be more and more studies to discover new biological functions of circRNAs in stem cells. Therefore, we would like to change the article title to “Circular RNAs acting as miRNAs ’sponges and their roles in stem cells”.

Q2: In the same logic, the abstract of the manuscript is not presenting the full content of the research. Biomarkers are very interesting in the clinical scenario, but from the data presented, I believe there is not enough data to provide the concept of a circRNA biomarker yet.

Response: Thanks for your suggestion. We agree with you that circRNAs may represent a new class of biomarkers for human disease. Therefore, we added the content of circRNA biomarkers in Abstract and Introduction sections.

Q3: Section 5 has only two references cited.

Response: Thanks for your suggestion. We have cited more references in Section 5.

Q4: Figure 1 should have a legend fully explaining its content. Figure 2 (which is excellent) should also be more explained.

Response: Thanks for your suggestions. We have added figure legends to explain Figure 1 and Figure 2 in the corresponding position. And in Figure 2, we added the role of circRNAs in regulating cancer stem cells.

Q5: The sentences in lines 101-105 are confusing. Please rephrase it.

Response: We have rephrased the sentences. The revised sentences are “miRNAs have shown to function as gene expression regulators in some diseases. They are post-transcriptional regulate gene expression via inhibit its translation or facilitate its degradation through direct base pairing to the 3’untranslated region (UTR), the coding region or the 5’-UTR of target mRNA. CircRNAs can function as miRNA sponges and act as competitive endogenous RNAs to deregulate mRNA by miRNA.”

Minor points:

Q6: The link for the TSCD database is not working.

Response: The TSCD database (http://gb.whu.edu.cn/TSCD) is working using our computers. If other users have any problems or questions, users can contact the developer Professor He at Wuhan University (Chunjiang He, Associate Professor, School of Basic Medical Sciences, Wuhan University, Email: che@whu.edu.cn).

Figure 1. The web interface of TSCD

Figure 2.  The distribution of global users

Q7: MBL protein should be first mentioned in the Introduction section, as QKI.

Response: Thanks for your suggestion. We introduced the splicing factors, MBL and QKI proteins, which regulate the production of circRNAs depending on the specific binding sequences in the second paragraph of the Introduction section.

Q8: I suggest the first paragraph of the Conclusions and Perspectives be moved to Section 4.

Response: Thanks for your suggestion. We have moved the first paragraph of the Conclusions and Perspectives to the begin of Section 4.

Reviewer 3 Report

The review manuscript by Juan Xiao et al. describes an attempt to summarize the recent data about the features of circular RNAs and their implications in stem cell biology. The work is well-organized and the logic flow of data presentation is clear. The manuscript is acceptable for publication; however, minor revision can make it more valuable. Thus, minor revision is recommended.

Minor issues

  1. It would be great if authors could describe the involvement of different circRNAs in physiological and pathological processes.
  2. In recent studies, it was shown that apoptotic endonuclease G (EndoG) can produce splice-switching oligonucleotides that act very similarly to circRNA. Such nucleotides (EGPO – EndoG-produced oligonucleotides) were disrobed for human Telomerase Reverse Transcriptase (hTERT) and Deoxiribonuclease I (DNase I). The authors should describe the functioning of EndoG during the production of circ-RNA-similar oligonucleotides.

Author Response

The review manuscript by Juan Xiao et al. describes an attempt to summarize the recent data about the features of circular RNAs and their implications in stem cell biology. The work is well-organized and the logic flow of data presentation is clear. The manuscript is acceptable for publication; however, minor revision can make it more valuable. Thus, minor revision is recommended.

Thanks for your encouragement and advice.

Minor issues

Q1: It would be great if authors could describe the involvement of different circRNAs in physiological and pathological processes.

Response: Thanks for your suggestions. We agree with you that different circRNAs play essential roles in physiological and pathological processes, such as aging, insulin secretion, tissue development, atherosclerotic vascular disease risk, cardiac hypertrophy and cancer. Therefore, we added relevant research in the Introduction section. Because we focused on the function of circRNAs in stem cells, so we only added those relevant examples in the Introduction section.

Q2: In recent studies, it was shown that apoptotic endonuclease G (EndoG) can produce splice-switching oligonucleotides that act very similarly to circRNA. Such nucleotides (EGPO – EndoG-produced oligonucleotides) were disrobed for human Telomerase Reverse Transcriptase (hTERT) and Deoxiribonuclease I (DNase I). The authors should describe the functioning of EndoG during the production of circ-RNA-similar oligonucleotides.

Response: Thanks for your suggestions. We carefully read the research progress about EndoG and hTERT, and discovered that the alternative splicing of hTERT plays crucial roles in various physiological and pathological processes, and mountains of research on hTERT and EndoG is very meaningful. Therefore, we think EndoG should be function in the production of circRNAs or other noncoding RNAs, considering this, researchers should pay attention to the studies of EndoG. Certainly, we added the research results to the corresponding position (Section 2.1 Biogenesis and types of circRNAs) and the relevant literature is cited (Zhdanov DD, Pokrovsky VS, Orlova EV, Orlova VS, Pokrovskaya MV, Aleksandrova SS, Sokolov NN: Intracellular Localization of Apoptotic Endonuclease EndoG and Splice-Variants of Telomerase Catalytic Subunit hTERT. Biochemistry (Mosc) 2017, 82(8):894-905.

Zhdanov DD, Vasina DA, Orlova VS, Gotovtseva VY, Bibikova MV, Pokrovsky VS, Pokrovskaya MV, Aleksandrova SS, Sokolov NN: Apoptotic endonuclease EndoG induces alternative splicing of telomerase catalytic subunit hTERT and death of tumor cells. Biomed Khim 2016, 62(3):239-250.

Plyasova AA, Zhdanov DD: Alternative Splicing of Human Telomerase Reverse Transcriptase (hTERT) and Its Implications in Physiological and Pathological Processes. Biomedicines 2021, 9(5).

Zhdanov DD, Vasina DA, Grachev VA, Orlova EV, Orlova VS, Pokrovskaya MV, Alexandrova SS, Sokolov NN: Alternative splicing of telomerase catalytic subunit hTERT generated by apoptotic endonuclease EndoG induces human CD4(+) T cell death. Eur J Cell Biol 2017, 96(7):653-664.).

Round 2

Reviewer 1 Report

The authors have incorporated the changes as suggested. I would recommend the acceptance  of this manuscript.